# *Narcissus* Plants: A Melting Pot of Potyviruses

**DOI:** 10.3390/v14030582

**Published:** 2022-03-11

**Authors:** Wiwit Probowati, Shusuke Kawakubo, Kazusato Ohshima

**Affiliations:** 1Laboratory of Plant Virology, Department of Biological Resource Science, Faculty of Agriculture, Saga University, 1-banchi, Honjo-machi, Saga 840-8502, Japan; 20976002@edu.cc.saga-u.ac.jp (W.P.); 20751004@edu.cc.saga-u.ac.jp (S.K.); 2The United Graduate School of Agricultural Sciences, Kagoshima University, 1-21-24 Korimoto, Kagoshima 890-0065, Japan

**Keywords:** *Narcissus* plants, co-infection, narcissus late season yellows virus, narcissus degeneration virus, potyvirus

## Abstract

Our paper presents detailed evolutionary analyses of narcissus viruses from wild and domesticated *Narcissus* plants in Japan. Narcissus late season yellows virus (NLSYV) and narcissus degeneration virus (NDV) are major viruses of *Narcissus* plants, causing serious disease outbreaks in Japan. In this study, we collected *Narcissus* plants showing mosaic or striped leaves along with asymptomatic plants in Japan for evolutionary analyses. Our findings show that (1) NLSYV is widely distributed, whereas the distribution of NDV is limited to the southwest parts of Japan; (2) the genomes of NLSYV isolates share nucleotide identities of around 82%, whereas those of NDV isolates are around 94%; (3) three novel recombination type patterns were found in NLSYV; (4) NLSYV comprises at least five distinct phylogenetic groups whereas NDV has two; and (5) infection with narcissus viruses often occur as co-infection with different viruses, different isolates of the same virus, and in the presence of quasispecies (mutant clouds) of the same virus in nature. Therefore, the wild and domesticated *Narcissus* plants in Japan are somewhat like a melting pot of potyviruses and other viruses.

## 1. Introduction

Studies of the co-infection of plant viruses in wild and domesticated plants are important for understanding plant virus epidemiology, evolution, and virus–virus interactions; consequently, these investigations will contribute to developing efficient and durable control strategies [1,2,3,4,5]. Co-infections by two or more plant viruses [6,7,8,9,10,11], different isolates of the same viruses [12,13], and the presence of quasispecies of the same viruses [12,14,15,16,17] in wild and domesticated plants seem likely to be common in nature [18,19,20]. Plants of the genus *Narcissus* in the family Amaryllidaceae are known as wild or domesticated plants, although it is difficult to draw a boundary distinguishing between these plants in Japan. *Narcissus* plants are also known to be infected with more than twenty different virus species [13,21,22,23,24,25,26,27,28]. For instance, many viruses that infect *Narcissus* plants belong to the genus *Potyvirus*; cyrtanthus elatus virus A (CyEVA), iris severe mosaic virus (ISMV), lily mottle virus, narcissus late season yellows virus (NLSYV), narcissus yellow stripe virus (NYSV), narcissus degeneration virus (NDV), Indian narcissus virus, and ornithogalum mosaic virus. Other major viruses known to infect wild and domesticated *Narcissus* plants worldwide are narcissus latent virus (NLV, macluravirus), narcissus symptomless virus (carlavirus), narcissus common latent virus (NCLV, carlavirus), nerine latent virus (NeLV, carlavirus) [13,27,28,29,30,31,32]. To date, 26 virus species and three tentative species were also reported to infect *Narcissus* plants but some of them were distributed in the limited countries of the world [29].

Viruses in the genus *Potyvirus* in the family *Potyviridae* are distributed worldwide and infect a wide range of both mono- and dicotyledonous plant species, including *Narcissus* plants [3,33,34]. They are transmitted non-persistently by many aphid species [35] and are flexuous filamentous particles 700–750 nm long, each of which contains a single copy of the genome. The genome is a single-stranded, positive-sense RNA of about 10,000 nucleotides with one open reading frame (ORF) that is translated into one large polyprotein which is known to be post-translationally cleaved by virus-encoded proteases, and with a small overlapping ORF named pretty interesting potyviridae ORF (PIPO) [3,36].

*Narcissus* plants are likely to harbor co-infections of potyviruses and other viruses in several areas of China, European countries, India, New Zealand, and Japan [13,22,23,31,37,38,39,40,41,42] but not in the sites surrounding the Perth metropolitan area in Western Australia [43]. Furthermore, there is a report that NLSYV was singly infected [39]. Co-infections of narcissus viruses have been reported on Kyushu Island in Japan [13], and whether and how they might be distributed in other parts of Japan is still largely unknown. Only a taxonomic study of NYSV-like viruses has been reported so far [28]. The term NYSV-like virus refers to the narcissus viruses in the turnip mosaic virus (TuMV) phylogenetic group that does not include NLSYV. Note that TuMV phylogenetic group includes NLSYV, NYSV, Japanese yam mosaic virus (JYMV), scallion mosaic virus (ScaMV) and TuMV [28].

In this study, we collected wild and domesticated *Narcissus* plants throughout Japan and screened for the presence of potyvirus infections using potyvirus-specific primers. The Japanese *Narcissus* plant (we call Nihon-zuisen, *Narcissus tazetta* var. chinensis) is known to differ from western *Narcissus* (daffodil). Furthermore, we focused on the distribution and evolution of narcissus potyviruses of NLSYV and NDV because these two appeared to be major viruses in Japan in the earlier study [13].

## 2. Materials and Methods

### 2.1. Narcissus Plant Samples

One-hundred eighty-nine wild and domesticated *Narcissus* plant leaves showing mosaic or chlorotic stripe and asymptomatic plant leaves were collected from different sites on the fields and banks of rivers in Japan, including home gardens and flower beds, during the winter and spring seasons of 2009–2015. Details of the plants and isolates, their place of origin, collection dates, host plants, and symptoms are shown in Appendix A.

### 2.2. Detection of NLSYV and NDV by RT-PCR and Sequencing

Because the sap from the collected *Narcissus* plant leaves in Japan did not induce local lesions on *Tetragonia tetragonioides* (formerly *T. expansa*), *Chenopodium amaranticolor*, *C. quinoa*, *Nicotiana benthamiana*, or *N. tabacum* plants, we could not clone the viruses biologically. Therefore, the viruses were directly identified from the sampled symptomatic or asymptomatic *Narcissus* plant leaves by reverse-transcription polymerase chain reaction (RT-PCR). We used potyvirus-specific primer pairs that were expected to amplify all potyviruses (Appendix A) and determined the partial nucleotide sequences of the cloned RT-PCR products as described below. Total RNA was extracted using Isogen RNA extraction reagent (Nippon Gene, Tokyo, Japan) from the plant leaves (Appendix A). We reverse-transcribed the viral RNAs using PrimeScript II Moloney murine leukemia virus reverse transcriptase (Takara Bio, Shiga, Japan), and amplified the potyvirus cDNAs using high-fidelity Prime STAR GXL DNA Polymerase with a PrimeScript II High Fidelity One-Step RT-PCR Kit (TaKaRa Bio, Shiga, Japan). The RT-PCR conditions were: 45 °C for 10 min for RT, one cycle of 94 °C for 2 min, and 40 cycles of 98 °C for 10 s, 45 °C for 15 s, and 68 °C for 35 s. RT-PCR products of 1850–2100 bp for NLSYV, NDV, and some other potyviruses were amplified from *Narcissus* plant leaves using potyvirus-specific primer pairs [13,28] (Appendix A), POTYNIbNOT4P (forward, 5′-GGGGCGGCCGCATATGGGGTGAGAGAGGTNTGYGTNGAYGAYTTYAAYAA-3′) and Tu3T9M (reverse, 5′-GGGGCGGCCGCT_15_-3′), or onion yellow dwarf virus (OYDV) phylogenetic group-specific primer pairs, RGNDNIBNOT4P (forward, 5′-GGGGCGGCCGCATATGGGGTGAGAGAGGRTAYSRWGGGAAGAAGAAGGA-3′) and Tu3T9M for virus genome amplification (underlined nucleotides; position of POTYNIB5P forward primer, see later). These primer pairs amplify most of the potyviruses and viruses in the OYDV phylogenetic group. The OYDV phylogenetic group consists of CyEVA, ISMV, OYDV, and shallot yellow stripe virus (SYSV). We separated the amplified cDNAs by electrophoresis in agarose gels and purified them using a QIAquick Gel Extraction Kit (Qiagen K.K., Tokyo, Japan). The nucleotide sequences of the amplified RT-PCR products from nuclear inclusion b (NIb) protein coding region to the 3′ end region (NIb-3′NCR region) from symptomatic and asymptomatic *Narcissus* plant leaves were firstly determined using a direct sequencing method. However, we often found multiple peaks at positions in the nucleotide sequence signals, indicating co-infections of different potyviruses, different isolates of the same virus, or the presence of quasispecies of the same virus in a single *Narcissus* plant. Therefore, only cloning was considered feasible for determining the genomic sequences of each virus in a single *Narcissus* plant. The RT-PCR products were cloned into the NotI site of plasmid pZErO-2 (Invitrogen, ThermoFisher Scientific, Tokyo, Japan). Three to 20 independent clones from each virus-infected *Narcissus* plant were obtained.

Firstly, the nucleotide sequences (600–700 bp) of parts of the NIb-3′NCR regions of all clones obtained from 120 isolates were first determined using POTYNIB5P forward primer (5′-CGCATATGGGGTGAGAGAGG-3′), a part of POTYNIbNOT4P or RGNDNIBNOT4P, using a BigDye Terminator v3.1 Cycle Sequencing Ready Reaction Kit (Applied Biosystems, Foster City, CA, USA) and an Applied Biosystems Genetic Analyzer DNA model 3130. Because large numbers of clones were obtained (Appendix A), we first aimed to identify the viruses in each *Narcissus* plant and selected clones using the 600–700 nucleotide sequences obtained by POTYNIB5P primer for further sequencing. As a result, BLAST searches showed that the cloned partial NIb and coat protein (CP) sequences are closely related to one or other of the sequences of potyviruses of CyEVA, NDV, NLSYV, OrMV, and NYSV, or a macluravirus of NLV. The sequences of NIb-3′NCR clones from the NLSYV and NDV genomes were also determined in both directions using the narcissus virus-specific primers (Appendix A) and primer walking. Sequence data were assembled using BioEdit version 5.0.9 [44].

### 2.3. Sequencing of Full Genomics of NLSYV and NDV

Since the information of consensus sequences in the internal genomic regions of NLSYV and NDV was insufficient to design primers, we applied the primer walking method to obtain the full genomic sequences of NLSYV and NDV isolates. The appropriate primers were designed based on the full genomic sequences of the TuMV or OYDV phylogenetic group viruses available from GenBank (Appendix A). Finally, three to five fragments (Appendix A) of RT-PCR products were used to obtain the full genomic sequences of NLSYV and NDV. The RT-PCR products were cloned into the NotI site of plasmid pZErO-2. We obtained at least three independent clones for each fragment. We overlapped the regions between RT-PCR products at least 450 nucleotides. The clones that had no mismatch in the overlapping regions were concatenated to obtain full genomic sequences. These overlapping fragments ensured no artificial recombination events in the NLSYV or NDV genomes during the sequencing and evolutionary analyses. The nucleotide sequences of clones were determined in both directions by primer walking using approximately 80 primers (Appendix A) because of variations between the sequences of each isolate. Finally, we determined the full genomic sequences of 25 NLSYV isolates and five NDV isolates.

### 2.4. Neighbor-Net Phylogenetic Network of CP Coding Regions

We inferred the phylogenetic network of NLSYV and NDV using Neighbor-net methods implemented in SPLITSTREE v4.11.3 [45]. A total of 248 CP coding nucleotide sequences of NLSYV isolates and 125 CP coding nucleotide sequences of NDV (Appendix A) obtained by the cloning methods were aligned with those of the same viruses available from GenBank with outgroup taxa. The outgroup included the CP coding sequences of the isolates of JYMV [46] (acc # KJ701427), TuMV [47] (acc # AB701690), NYSV [26] (acc # JQ911732), and ScaMV [48] (acc # NC_003399). For NLSYV, the outgroup included those of the isolates of CyEVA [27] (acc # NC_017977), ISMV [49] (acc # NC_029076), OYDV [50] (acc # JX433020), and SYSV [51] (acc # AM267479) for NDV. The alignments were made using CLUSTAL_X2 [52] with TRANSALIGN [53] to maintain the degapped alignment of the encoded amino acids. The aligned nucleotide sequences of CP coding regions used to infer the phylogenetic network of NLSYV with TuMV phylogenetic group viruses were 810 nucleotides in length, whereas those of NDV with OYDV group viruses were 750 nucleotides.

### 2.5. Maximum Likelihood Phylogenetic Tree of Polyprotein Coding Regions

A total of 25 polyprotein coding nucleotide sequences of NLSYV isolates determined in this study with those of three sequences from GenBank (Marijiniup8 acc # NC_02362, Marijiniup9 acc # JX156421, and Zhangzhou acc # JQ326210), and five polyprotein coding sequences of NDV determined in this study with those of two sequences from GenBank (Zhangzhou acc # NC_008824 and Marijiniup2 acc # JQ395041) were used to construct maximum likelihood phylogenetic trees.

The isolates of JYMV [46] (acc # KJ701427), TuMV [47] (acc # AB701690), NYSV [23,26,27] (acc # NC_011541, JQ911732 and JQ395042), and ScaMV [48] (acc # NC_003399) were used as outgroup taxa for NLSYV polyprotein coding sequences, and the isolates of CyEVA [27] (acc # NC_017977), ISMV [49] (acc # NC_029076), OYDV [50] (acc # JX433020), and SYSV [51] (acc # AM267479) were used as outgroup taxa for NDV polyprotein coding sequences. The alignments were performed using CLUSTAL_X2 [52] with TRANSALIGN [53] to maintain the degapped alignment of the encoded amino acids. The aligned sequences of the polyprotein coding regions were 8970 nucleotides for NLSYV and 9093 nucleotides for NDV after removing gaps. Note that incomplete nucleotide sequences with ambiguous nucleotides from GenBank were not included in the analyses.

Partial regions of polyprotein coding sequence (nt 2076-7471) were used for NLSYV maximum likelihood analysis because the recombination sites were identified at positions around nt 2000 and nt 7500 (see later). Conversely, the complete polyprotein coding sequences were used for NDV maximum likelihood analysis because no evidence of recombination was found.

Phylogenetic relationships were inferred using the maximum likelihood method implemented in PhyML version 3.1 [54] under the general time-reversible substitution model with a proportion of invariable sites and gamma-distributed site rates (GTR+I+Γ4). We used jModeltest version 2.1.2 [55] to determine the best-fit model of nucleotide substitution for each dataset. Branch support was evaluated by the bootstrap method based on 1000 pseudoreplicates. The inferred trees were visualized using TREEVIEW [56].

### 2.6. Recombination Analysis

We reassembled the aligned 5′ and 3′ NCR sequences with both ends of the aligned NLSYV and NDV polyprotein coding sequences to form nearly complete genomic sequences of 9556 nucleotides for NLSYV and 9786 nucleotides for NDV, excluding the 24 nucleotides that were used to design the primer for RT-PCR amplification. The nearly complete genomic sequences were assessed for evidence of recombination, especially for recombination sites in both polyproteins and in the NCRs. Putative recombination sites in all sequences were identified using RDP [57], GENECONV [58], BOOTSCAN [59], MAXCHI [60], CHIMAERA [61], and SISCAN [62] programs implemented in the RDP4 version 101 software package [63] and the original SISCAN version 2 program [62]. First, the incongruent relationships were checked using the programs implemented in RDP4. These analyses were performed using default settings (except to use ‘sequences are linear’) for the different detection programs and a Bonferroni-corrected p-value cutoff of 0.01. All isolates that were identified as likely recombinants by the programs in RDP4, supported by three different methods with an associated *p*-value of <1.0 × 10^−4^ (i.e., the most likely recombination sites), were re-checked using the original SISCAN version 2 with all nucleotide sites. We checked all sequences with a 100 and 50 nucleotide sliding window for evidence of recombination using the SISCAN program. These analyses also assessed which non-recombinant sequences had regions that were closest to those from the recombinant sequences, indicating the likely lineages of the donor sequence that provided those regions of the recombinant genomes. For convenience, we called them the “parental isolates” of recombinants. Second, we also aligned NLSYV or NDV sequences without outgroup taxa sequences and directly checked for evidence of recombination using the programs. Finally, GARD [64] was used to assess sites for any evidence of recombination.

### 2.7. Genetic Diversity

The nucleotide diversities and identities of 28 NLSYV and seven NDV were estimated using MEGA version 7 [65], EMBOSS Needle [66], and the Sequence Demarcation Tool (SDT) v1.2 [67]. We evaluated the degree of mutational saturation in the polyprotein coding sequences using the substitution saturation (Iss) statistic in DAMBE version 6.4.81 [68].

### 2.8. Spatial Diffusion of NLSYV and NDV

We applied a continuous phylogeographic model [69] to reconstruct the spatial diffusion of NLSYV and NDV in Japan. We used 25 NLSYV isolates of recombination-free partial polyprotein coding sequence (nt 2076–7471) and five NDV isolates of complete polyprotein coding sequence after removing isolates sampled outside Japan. The polyprotein coding sequences of each isolate were assigned two-dimensional geographic coordinates (i.e., latitude and longitude) (Appendix A) and analyzed using BEAST v1.10.4 [70]. The posterior distribution of each parameter was inferred based on the Markov chain Monte Carlo run for 100 million steps each, sampled every 10,000 steps. Tracer v1.7.1 [71] was used to check for convergence and satisfactory mixing, based on the effective sample size exceeding 200 for each parameter. We carried out the phylogeographic reconstruction using the gamma-relaxed random walk model that was supported by marginal likelihood estimation based on path sampling and stepping-stone sampling methods [72]. The 95% credible intervals area of the inferred location for each lineage was visualized using SpreaD3 [73] based on 1000 subsampled trees after removing burn-in states.

### 2.9. Detection of Carlaviruses

We inoculated NLSYV and NDV, NCLV, and NeLV-free asymptomatic *Narcissus* plants with the saps of *Narcissus* plant leaves infected with NLSYV (NY-HO42, NY-A65, and NY-HR39 plants) or NDV (NY-KG11 and NY-FI23 plants). We inoculated the virus-infected saps when the *Narcissus* plant leaves were germinated and very small, namely before the leaves expand in the winter and spring season. The potyvirus- and carlavirus-free *Narcissus* plants were confirmed for the presence or absence of potyviruses and carlaviruses using the narcissus potyvirus- and carlavirus-specific primer pairs (Appendix A). For calravirus detection, we used NCLV-specific forward primer, CARNCLVCP4P, 5′-GCGGCCGCCTGACCCCAGCAATCCTTACAA-3′ or NeLV-specific forward primer, CARNLVCP5P, 5′-GCGGCCGCARAARGGKTGGAGRCCTTCYTC-3′ and carlavirus-specific reverse primer Tu3T9M for amplifying the RT-PCR products of the CP coding regions of carlaviruses and used NCLV-specific forward primer CARNCLVCP1P, NeLV-specific forward primer CARNLVCP2P, and NCLV and NeLV-specific forward primer CARNACP3P forward primers for their sequencing. We applied RT-PCR, cloning, and sequencing methods similarly to for narcissus potyviruses. We focused on the carlaviruses of NCLV and NeLV because they were important narcissus viruses detected in *Narcissus* plants in the earlier studies [27,74].

## 3. Results and Discussion

### 3.1. Co- and Single-Infection of Narcissus Viruses

We determined parts of nucleotide sequences (approximately 600–700 bp) in the NIb protein-coding regions of 1174 clones obtained from 189 symptomatic and asymptomatic *Narcissus* plant leaves (Appendix A) using POTYNIB5P primer. Of the 189 plants, 120 (64%) were infected with NLSYV, NDV, NYSV, and other narcissus viruses of the genus *Potyvirus* and *Macluravirus* in the family *Potyviridae* (Figure 1, Table 1). Of 120 *Narcissus* plants, 12 (10%) were co-infected with three viruses and all were co-infected with NLSYV and NYSV. We found that 36 (30%) plants were co-infected with two different viruses, and 15 (13%) were co-infected with NLSYV and NYSV. Therefore, 48 (40%) plants in total were co-infected. A total of 72 (60%) plants were singly infected, and 37 (31%), 23 (19%), and 3 (3%) plants were infected with NLSYV, NYSV, and NDV, respectively; therefore, the major narcissus potyviruses in Japan are NLSYV, NYSV, and NDV. We focused on the distribution and evolution of NLSYV and NDV in this study because the epidemiology of NYSV-like viruses has already been reported in an earlier study [28]. Note that no RT-PCR product was obtained from the amplification by potyvirus-specific primers of 69 *Narcissus* plants, even though many showed virus-like symptoms (Appendix A). This indicates that these *Narcissus* plants were infected with viruses that do not belong to the *Potyviridae* family.

### 3.2. 3′ Terminal Region Sequences

We determined the remaining nucleotide sequences of the NIb-3′NCR regions of 248 clones of NLSYV and 125 clones of NDV (Appendix A). A total of approximately 658,000 nucleotides were sequenced. The nucleotide sequences of the NIb-3′NCR regions of each NLSYV and NDV clone determined in this study were approximately 2250 nucleotides in length.

The CP coding regions of NLSYV reported from Australia and China and those sequenced from Japan in this study were all 822 nucleotides (274 amino acids) in length. By contrast, the length of 3′NCR varied: Australian Marijiniup8 and Marijiniup9 isolates (acc # NC_023628 and JX156421, respectively) were 289 and 158 nucleotides in length, respectively; the Chinese Zhangzhou isolate (acc # JQ326210) and all 248 clones from the 25 Japanese NLSYV isolates were 210 nucleotides in length. Conversely, the CP coding regions and the 3′ NCR of NDV reported from all the countries including Japan were all 780–783 nucleotides (260–261 amino acids) and 148 nucleotides in length, respectively.

### 3.3. Phylogenetic Relationships and Nucleotide Identities Assessed by Coat Protein Coding Sequences

We inferred the phylogenetic network using 248 NLSYV and 125 NDV CP coding sequences. NLSYV CP coding sequences fell into at least five major clades in the phylogenetic network (Figure 2A), whereas the NDV CP coding sequences fell into two major clades (Figure 2B). Therefore, we chose 25 NLSYV and 5 NDV isolates for subsequent full genomic sequencing, so that they represented all the major clades found in the CP phylogenetic network.

### 3.4. Co-Infection with Different Isolates and Quasispecies

One of the difficulties in this study was obtaining the full genomic sequences of NLSYV and NDV using several cloned plasmids that cover partial genomic regions because we were unable to isolate either virus from single lesion infections. Therefore, we needed to characterize the sequences from a single RNA molecule in the infected *Narcissus* plants. We amplified the longest possible RT-PCR products with the aim of obtaining full genomic sequences from the cloned fragments with no mismatches in the overlapping regions (Appendix A). As a result, the full genomic sequences of NLSYV and NDV isolates were obtained from three to five overlapping clone fragments. We found many nucleotide mismatches among the cloned fragment sequences, and a single *Narcissus* plant was co-infected with different isolates of the same virus or quasispecies of the same virus (Appendix A).

For instance, among 23 NLSYV isolates, five isolates (NY-CB3, NY-CB4, NY-KW4, NY-M4, and NY-FK266) were co-infected with different isolates (65–300 mismatches between the clones), 16 were co-infected with quasispecies (1–17 mismatches between the clones), and only two isolates (NY-F1 and NY-AC230) had no mixed nucleotides between the overlapping fragment clones. Similarly, we found many nucleotide mismatches between the sequences of NDV overlapping fragments. Narcissus are bulbous plants, and narcissus potyviruses are transmitted by aphids; hence, the plants have numerous chances to be co-infected [13,75]. Note that we used high-fidelity Taq DNA polymerase for amplifying RT-PCR products. The fidelity of the polymerase was also checked, as potentially contributing to the observed mismatches, by amplifying and then sequencing 12,000 nucleotides of PCR products from cloned cDNA of known sequence [15], and no mismatches were found compared with the original sequences. Finally, we found clones with no mismatches in the overlapping regions of each NLSYV and NDV isolate and successfully obtained the full genomic sequences of all the isolates from a single RNA molecule but not from a different RNA molecule of the same virus in the infected Narcissus plants.

### 3.5. Full Genomic Sequences

The NLSYV genomes of Japanese isolates were 9625–9628 nucleotides in length, excluding the 5′ end 24 nucleotide primer sequences. All 25 Japanese isolates of the polyprotein coding regions were 9315 nucleotides with 3105 amino acids in length. The lengths of 5′ and 3′ NCRs of NLSYV were 99 nucleotides, excluding the 5′ end primer sequences and 211–214 nucleotides, respectively. The lengths of encoding protein 1 (P1), helper-components proteinase (HC-Pro), P3, overlapping ORF (PIPO), 6K1, CI, 6Kda 2 protein (6K2), genome-linked viral protein (VPg), NIa-Pro, NIb, and CP proteins were 957, 1374, 1062, 195, 156, 1932, 159, 573, 729, 1551, and 822 nucleotides, respectively. Therefore, the lengths of the genomes and protein-coding regions are similar to those of NYSV and NYSV-like isolates, which belong to the TuMV phylogenetic group [28]. The lengths of P1 and P3 varied among narcissus viruses in the TuMV and OYDV phylogenetic groups. All the motifs reported for different potyvirus encoded proteins were found. The CP region of NY-CB4 and NY-OI4 clones (Appendix A) belonged to NLSYV Clades 2 and 4 respectively (Figure 2A), whereas all the full and partial genomic sequences including the CP region of both isolates belong to Group 1 after we obtained the full genomic sequences (Appendix A). This indicates that the full genomic sequence of NY-CB4 and NY-OI4 was obtained from the different viral populations from the CP sequences in Clades 2 and 4 of both plants.

The NDV genomes of Japanese isolates were 9789–9792 nucleotides in length, excluding the 5′-end 24 nucleotide primer sequences. Five Japanese isolates of polyprotein coding regions were 9552–9555 nucleotides with 3184–3185 amino acids in length. The lengths of 5′ and 3′ NCRs of NDV were 89 nucleotides, excluding the 5′-end 24 nucleotide primer sequences, and 148 nucleotides. The lengths of encoding P1, HC-Pro, P3, overlapping ORF (PIPO), 6K1, CI, 6K2, VPg, NIa-Pro, NIb, and CP proteins were 1182, 1374, 1152, 234, 156, 1908, 159, 555, 723, 1560, and 780–783 nucleotides, respectively. The nucleotide sequence data reported in this paper are available in the GenBank/ENA/DDBJ database as accession LC664177-LC664181 for NDV isolates and LC664182-LC664206 for NLSYV isolates.

### 3.6. Inference of Recombination

The aligned polyprotein coding sequences of NLSYV were 8970 nucleotides in length. The phylogenetic network of the polyprotein coding sequences showed reticulated phylogenetic relationships, reflecting conflicts in the phylogenetic signal that are possibly due to the presence of recombinant sequences (data not shown). Recombination is an important source of genetic variation for potyviruses [3]. The 25 polyprotein coding sequences determined in this study, together with three sequences (Marijiniup8 acc # NC_02362, Marijiniup9 acc # JX156421, and Zhangzhou acc # JQ326210) obtained from GenBank, were assessed for evidence of recombination. Three recombination sites were identified by RDP4 in the NLSYV genomes: one located at the middle of the HC-Pro coding region (genome position around nt 2000) and the other two located in the CP coding region (Figure 3). Two (around nt 2000 and nt 7500) of three recombination sites were clearly supported by more than five programs in RDP4 (Appendix A). The recombination site in the HC-Pro coding region was found in both Japanese and Australian isolates. The recombination site at positions around nt 9000 was found in both Japanese and Australian NLSYV, but both seemed to be tentative, and the different parental sequences were identified by RDP4. On the other hand, no identical recombination type pattern of the genome was found between these three countries. Moreover, identical non-recombinants were found in China and Japan, and in Australia and Japan, but not in Australia and China. Fewer recombination sites were found in NLSYV compared to other potyvirus species [76,77]. However, no evidence of recombination sites was found in the NDV genome. The genomes of NDV isolates shared high nucleotide identity throughout their genomes. Therefore, we cannot exclude the possibility of recombination in NDV genomes, awaiting the discovery of supporting evidence using a large number of isolates collected over the world. We also used GARD [64] software to confirm the evidence of recombination sites (Figure 3B). The results of NLSYV also showed evidence of recombination at almost the same recombination sites as detected by RDP4. We also inferred phylogenetic trees using the regions between the recombination sites (Appendix A); 5′ end–nt 1800, nt 2200–7400, and nt 7700–3′end. The topology of the trees clearly indicated that there might be recombination sites between these genomic regions. Similarly, none of the recombination sites were found in NDV genomes. A more detailed collection of Narcissus plants from around the world would clarify the evidence of common and different recombination sites in the NLSYV and NDV genomes.

We compared the number of clades obtained by CP coding regions in the NLSYV phylogenetic network (Figure 2A) and that of groups in non-recombinant and recombinant genomes (Figure 3 and Appendix A). As a result, Clade 2 split into Group 2 and recombinant group (NY-F1, NY-FK266, NY-HG25, NY-HR39, and NY-OS1), and Clade 4 split into Groups 4 and 5. The NY-CB1 isolate fell into the independent clade (Clade 5) in the CP phylogenetic network and has a recombination site in CP coding region with Group 1 and 5 parents.

### 3.7. Inference of Phylogenetic Relationships and Nucleotide Diversities Assessed by Polyprotein Coding Sequences

The NLSYV partial polyprotein coding sequences (genome position nt 2076–7471) fell into five major phylogenetic groups (Figure 4A), and this grouping was also supported by pairwise identity comparisons by SDT v1.2. Phylogenetic analysis of the NLSYV polyprotein coding region shows that NLSYV has at least five phylogenetic groups. The groupings are also supported by full and partial genomic sequences of NLSYV (Appendix A). The grouping of isolates did not correspond well to their geographical origins. The pairwise identities of NLSYV sequences are higher than 81%. There are two clusters for NDV supported by high bootstrap values. Therefore, NDV polyprotein coding sequences seem to form two groups (Figure 4B) and the pairwise identities of NDV sequences are higher than 95%. Although we still need additional collections of NDV isolates from around the world to assess the diversity of the NDV population, our results indicate that in Japan the genetic diversities of NLSYV are higher than those of NDV.

The nucleotide sequence identities of complete polyprotein coding sequences of 28 NLSYV (9294–9315 nucleotides) and 7 NDV (9552–9555 nucleotides) were also calculated by EMBOSS Needle. The nucleotide identities of polyprotein coding sequences between each isolate are higher than 82% for NLSYV and higher than 94% for NDV. The identities are similar in most of the protein coding regions, whereas the P1 protein coding region is the most diverse, as previously reported for some potyviruses [4,78,79]. The nucleotide identities of different narcissus viruses are different: NLSYV, NYSV [28], and CyEVA [27,80] are genetically diverse, whereas NDV is the least variable. NYSV was first reported in 1908 in the UK [81], whereas NLSYV was first reported in *N. pseudonarcissus* in 1988 from the UK [39]. CyEVA, as a vallota speciosa virus, was first reported in 1980 from *Vallota speciosa* in The Netherlands [82,83], and NDV was first reported in *N. tazetta* in 1970 from the UK [84]. NLV is a species of macluravirus first reported in *N. pseudonarcissus* from the UK in 1967 [85]. Since NYSV, NLSYV, CyEVA, and NDV have narrow host ranges and are mostly limited to *Amaryllidaceae* host plants, as NLSYV has been reported from *Sternbergia lutea*, *Clivia miniata*, and NDV from *Lycoris*, the extent of nucleotide diversity might be related to the time of emergences of each virus. However, to validate this hypothesis, we need to compare them using the viruses collected from over the world.

### 3.8. Timescale Analysis, Migration of Non-Recombinants and Recombinants

We evaluated the degree of mutational saturation for NLSYV and NDV populations using Iss statistics. The estimated Iss was six to 23 times lower than the critical Iss value (Iss.c) for all datasets (*p* < 0.05). Therefore, there is little saturation across the NLSYV and NDV sequences in the datasets of each polyprotein coding region. We then attempted to infer the evolutionary timescale for NLSYV and NDV. However, a lack of temporal signals was suggested by the date-randomization test (data not shown) in both the NLSYV and NDV datasets of some genomic regions (e.g., HC-Pro, P3, NIb, and CP) and polyprotein coding regions. More serial collections of NLSYV and NDV isolates would allow us to measurably observe the evolving population.

The reconstructed spatial diffusion of NLSYV and NDV isolates in Japan were examined using the polyprotein coding region (Figure 5). The inferred distribution areas of NLSYV and NDV seem different in terms of the range of statistical uncertainty, even though NLSYV and NDV were often found in the same host plant as co-infections. The distributed area of statistical uncertainty for NLSYV is broader (Figure 5A), whereas that of NDV is more limited (Figure 5B), although both exist in the southwest parts of Japan. This is possibly due to the different periods of introduction into Japan for NLSYV and NDV. NDV might have been more recently introduced into Japan than NLSYV, considering the difference of the expanding area and nucleotide diversities of the two viruses collected in Japan. However, we still need further sampling of NLSYV and NDV both in Japan and neighboring countries to determine their time of introduction into Japan.

Our findings also raise an additional question as to why co- or single infection of viruses occurs in different host plants. For instance, our earlier studies of ScaMV from wild Japanese garlic plants (Chinese garlic or no-biru, a species of wild onion, *Allium macrostemon* Bunge) [77,86] showed almost no co-infection with other potyviruses, no co-infection with different viruses of the same isolates, and even the absence of quasispecies of the same virus. Japanese garlic and *Narcissus* are both monocotyledonous plants, and the infected potyviruses seem to have a narrow host range. The origins of domesticated populations of *Narcissus* are still unclear. They appear to have arisen in the area of the Iberian Peninsula, southern France, and northwestern Italy [87]. The Japanese *Narcissus* plant (Nihon-zuisen, *Narcissus tazetta* var. *chinensis*) was first described in the kanji dictionary as *Kagakushu*, which was published in 1444 CE in Japan. It is speculated that Japanese *Narcissus* plants drifted along oceanic currents from the Chinese continent to Japan before the Muromachi period (1336–1573 CE) at least three times or could have been introduced to Japan through trade with China, but the introduction of *Narcissus* plants to Japan is still unclear (Echizen Town, Ota Museum of Culture History, https://www.town.echizen.fukui.jp/otabunreki/ accessed on 10 January 2022). It is also speculated that *Narcissus* plants dispersed from Mediterranean countries via Silk Road to Eastern China [88]. Therefore, the *Narcissus* plant has a relatively long history in Japan. There are three main colonies of wild Japanese *Narcissus* plants: Echizen Seacoast (Fukui Prefecture), Kyonan-machi (Chiba Prefecture), and Nadakuroiwa (Hyogo Prefecture). The *Narcissus* plants *Narcissus poeticus* and *Narcissus jonquilla* were introduced from European countries to Japan after Meiji-Ishin (1868 CE). Japanese garlic plants are considered to be naturally part of the country’s flora. The epidemiological and evolutionary studies of co-infections of plant viruses have only recently started because it is difficult to collect co-infected plants from a wide area, so more such studies are needed in the near future [11,20,89].

### 3.9. Co-Infection with Carlaviruses

The potyvirus-free *Narcissus* plant leaves inoculated with the sap of NY-A65, NY-HO42, and NY-HR49 plants (infected with NLSYV) or NY-KG11 and NY-FI23 plants (infected with NDV) (Appendix A) showed chlorotic stripes and mosaic symptoms on *Narcisuss* (*N. tazzeta* var. *chinensis*) plant leaves one-month post-inoculation. The presence of NLSYV, NDV, NCLV, and NeLV in the inoculated plants was confirmed by RT-PCR amplification and sequencing. The *Narcissus* plants inoculated with the sap of NY-A65, NY-HO42, NY-HR49, NY-KG11, and NY-FI23 plant leaves were found to be infected with not only NLSYV or NDV but also NCLV. Therefore, many *Narcissus* plants in nature seem to be co-infected with not only potyviruses but also with carlaviruses [13,23,27,38,90]. A more detailed global analysis of narcissus carlaviruses will be reported in a future publication.

## 4. Conclusions

We presented a detailed evolutionary investigation of co-infections of narcissus viruses from wild and domesticated *Narcissus* plants in Japan. Our findings are that (1) NLSYV is widely distributed whereas the distribution of NDV is limited to the southwest parts of Japan; (2) the nucleotide sequence identities of the genomes between NLSYV isolates are higher than 82%, whereas those between NDV isolates are higher than 94%; (3) three novel recombination type patterns were found in the NLSYV population; (4) NLSYV has at least five distinct phylogenetic groups whereas NDV has two; and (5) infection with narcissus viruses often occur as co-infections of different viruses, different isolates of the same viruses, and in the presence of quasispecies (mutant clouds) of the same virus in nature. Our findings also illustrate that it is important to understand the co-infections of potyviruses in nature when designing control strategies in the future, as the wild and domesticated *Narcissus* plants in Japan are somewhat like a melting pot of potyviruses.

## Figures and Tables

**Figure 1 viruses-14-00582-f001:**
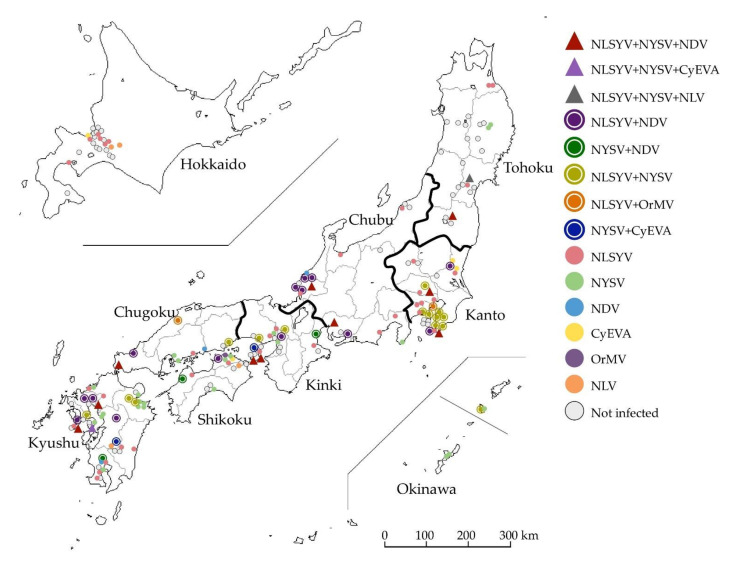
Collection map of wild and domesticated *Narcissus* plants in Japan. The *Narcissus* plants showing mosaic and chlorotic stripe and asymptomatic plants were collected from different sites on the banks of rivers and fields in Japan, including home gardens and flower beds, during the winter and spring seasons of 2009–2015. Color and shape of point on the map indicate the combination of detected virus, as shown in Table 1 and Appendix A. Cyrtanthus elatus virus A (CyEVA), narcissus degeneration virus (NDV), narcissus late season yellows virus (NLSYV), narcissus latent virus (NLV), narcissus yellow stripe virus (NYSV), and ornithogalum mosaic virus (OrMV). The map was obtained from http://www.craftmap.box-i.net/ (accessed on 10 January 2022).

**Figure 2 viruses-14-00582-f002:**
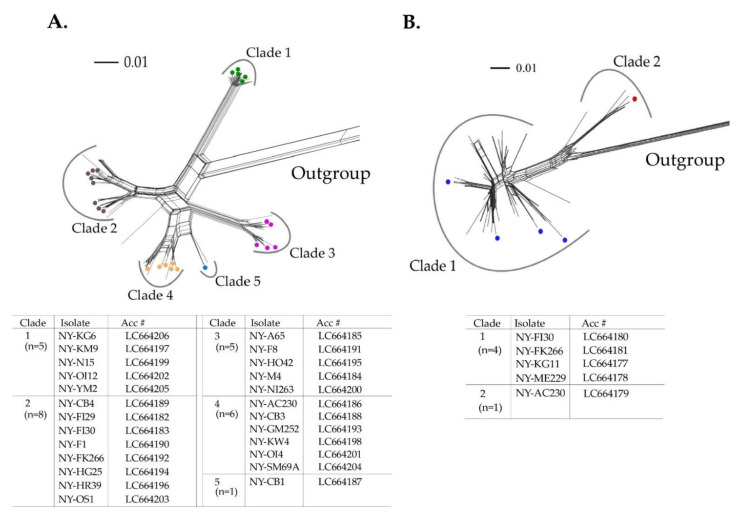
Phylogenetic networks of the nucleotide sequences of the complete coat protein (CP) coding regions from the plasmid clones of narcissus viruses. (**A**) Narcissus late season yellows virus (NLSYV). The nucleotide sequences of CP coding regions of Japanese yam mosaic virus (JYMV) [46], narcissus yellow stripe virus (NYSV) [26], scallion mosaic virus (ScaMV) [48], and turnip mosaic virus (TuMV) [47] in the TuMV phylogenetic group were used as outgroup taxa. (**B**) Narcissus degeneration virus (NDV). The nucleotide sequences of CP coding regions of cyrtanthus elatus virus A (CyEVA) [27], iris severe mosaic virus (ISMV) [49], onion yellow dwarf virus (OYDV) [50], and shallot yellow stripe virus (SYSV) [51] in the OYDV phylogenetic group were used as outgroup taxa. Dots in each clade show the isolates whose genomic sequences were fully sequenced. The full genomic sequences of the isolates determined in this study are listed in the tables below the phylogenetic networks of NLSYV and NDV. Acc # are also listed in the tables. The NY-CB1 isolate was found to have recombination site in CP coding sequence (see Section 3.6. Inference of Recombination).

**Figure 3 viruses-14-00582-f003:**
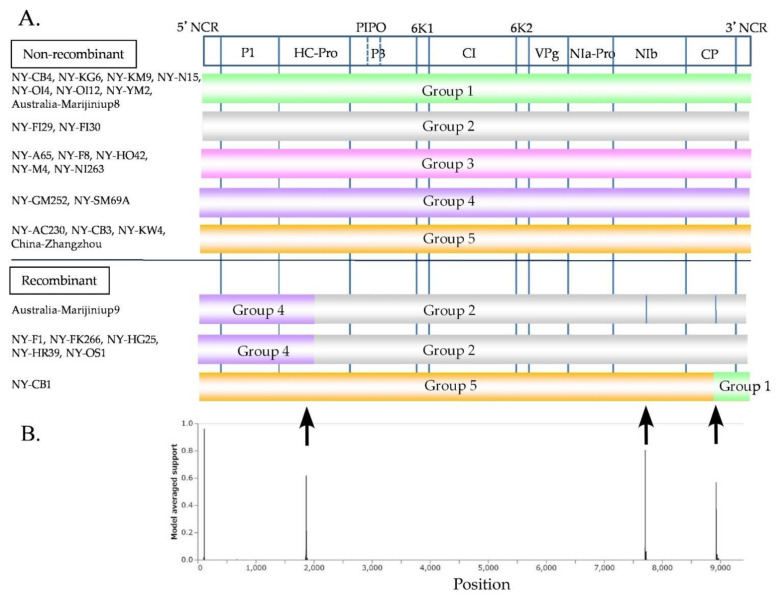
Recombination analysis. (**A**) Recombination genome map of the narcissus late season yellows virus (NLSYV) genomes of each isolate. Vertical solid lines show estimated approximate recombination sites. (**B**) The best placement of recombination sites (breakpoints) inferred by the algorithm for each site considered by GARD. The nucleotide positions correspond to the genome of the Zhangzhou isolate [26]. Note that recombination site around nt 9000 in Marijiniup9 genome is tentative.

**Figure 4 viruses-14-00582-f004:**
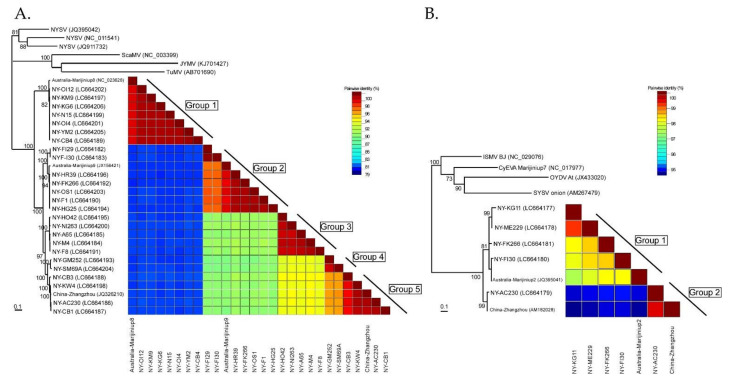
Phylogenetic relationships and pairwise identities of two narcissus viruses. (**A**) Twenty-eight partial polyprotein coding sequences (genome position nt 2076–7471) of narcissus late season yellows virus (NLSYV) were used to infer maximum likelihood phylogenetic trees. The polyprotein coding sequences of Japanese yam mosaic virus (JYMV) [46], narcissus yellow stripe virus (NYSV) [23,26,27], scallion mosaic virus (ScaMV) [48], and turnip mosaic virus (TuMV) [47] in the TuMV phylogenetic group were used as outgroup taxa. (**B**) Seven complete polyprotein coding sequences of narcissus degeneration virus (NDV) were used to infer maximum likelihood phylogenetic trees. The polyprotein coding sequences of cyrtanthus elatus virus A (CyEVA) [27] (acc # NC_017977), iris severe mosaic virus (ISMV) [49] (acc # NC_029076), onion yellow dwarf virus (OYDV) [50] (acc # JX433020), and shallot yellow stripe virus (SYSV) [51] (acc # NC_007433) in the OYDV phylogenetic group were used as outgroup taxa for NDV. Number at each node indicate bootstrap percentages based on 1000 pseudoreplicates for both trees and the values >70% are only shown.

**Figure 5 viruses-14-00582-f005:**
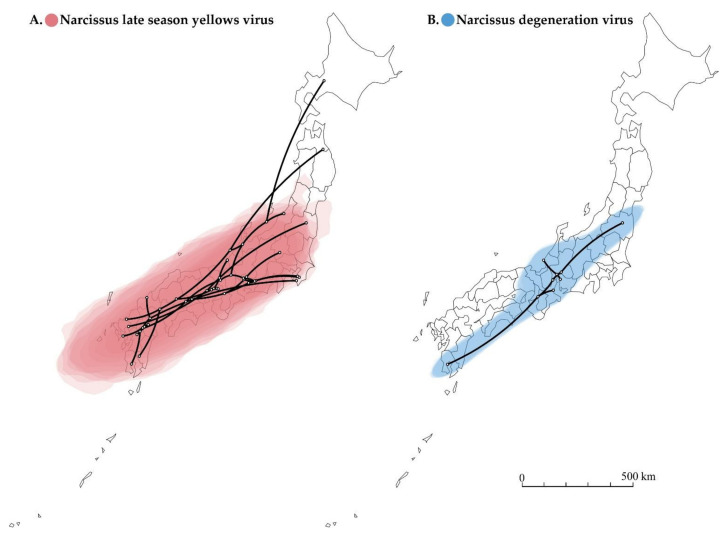
Spatial diffusion of narcissus viruses. (**A**) Twenty-five partial polyprotein coding sequences (genome position nt 2076–7471) of narcissus late season yellows virus (NLSYV). (**B**) Five complete polyprotein coding sequences of narcissus degeneration virus (NDV) were used. The points indicate the location of ancestral internal nodes and external tips. The 95% areas of credible intervals based on 1000 trees subsampled from the post-burn-in posterior distribution are shown as colored shadows. The map was obtained from https://gist.github.com/minikomi/4043986 (accessed on 10 January 2022).

**Table 1 viruses-14-00582-t001:** Detection of viruses in the family *Potyviridae*, including narcissus degeneration virus and narcissus late season yellows virus, in Japan.

			Numbers of Plants (%, Number of Plants Detected out of 120 Detected Plants)													
District (Island)	Plants Examined (n)	Plants Detected (n) (%) ^1^	Co-Infected with Three Viruses			Co-Infected with Two Viruses					Singly Infected with					
											Potyvirus ^2^					Maclura-Virus
											TuMV Group		OYDV Group			
			NLSYV NYSV NDV	NLSYV NYSV CyEVA	NLSYV NYSV NLV	NDV NLSYV	NDV NYSV	NLSYV NYSV	NLSYV OrMV	CyEVA NYSV	NLSYV	NYSV	NDV	CyEVA	OrMV	NLV
Hokkaido	24	9 (37.5)	0	0	0	0	0	0	0	0	6	0	0	1	0	2
Tohoku	25	7 (28.0)	1	0	1	0	0	0	0	0	3	2	0	0	0	0
Kanto	30	23 (76.7)	2	0	0	2	0	8	1	0	8	0	0	2	0	0
Chubu	19	14 (73.7)	2	0	0	5	0	0	0	0	5	1	1	0	0	0
Kinki	21	13 (61.9)	2	0	0	1	1	2	0	1	4	2	0	0	0	0
Chugoku	9	8 (88.9)	1	0	0	1	0	1	1	0	1	2	1	0	0	0
Shikoku	15	10 (66.7)	0	0	0	1	1	0	0	0	2	3	0	1	1	1
Kyushu and Okinawa	46	36 (78.3)	2	1	0	4	1	4	0	1	8	13	1	0	0	1
Total	189	120 (63.5)	10 (8.3)	1 (0.8)	1 (0.8)	14 (11.7)	3 (2.5)	15 (12.5)	2 (1.7)	2 (1.7)	37 (30.8)	23 (19.2)	3 (2.5)	4 (3.3)	1 (0.8)	4 (3.3)

^1^ This includes cyrtanthus elatus virus A (CyEVA), narcissus degeneration virus (NDV), narcissus late season yellows virus (NLSYV), narcissus latent virus (NLV), narcissus yellow stripe virus (NYSV), and/or ornithogalum mosaic virus (OrMV). Percentage (%); number of plants detected/number of plants examined for virus infection in each district. ^2^ TuMV group: turnip mosaic virus phylogenetic group and OYDV group: onion yellow dwarf virus phylogenetic group.

## Data Availability

All data obtained in this research is available upon request.

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
