# Peer review of "Narcissus Plants: A Melting Pot of Potyviruses"

_viruses, 2022, doi:10.3390/v14030582_

Round 1

Reviewer 1 Report

In this manuscript of an original article the authors collected 189 daffodil leaf samples of wild and domesticated plants in a 7-year period. They have found that 120 of the molecularly tested plants were infected with one or more members of the genus Potyvirus. Molecular testing, direct sequencing and later cloning and sequencing of this many samples is quite a job, tough in scientific circles it is expected that at least 2 different detection methods are used to verify the infection of a host. A possible second method could be ELISA testing, as it is relatively cheap and sensitive and can be upscaled, moreover there have been adequate number of reports using potyvirus group specific antibodies in bulbous ornamentals (including daffodils).

The title of the MS should be changed.

Firstly, Narcissus is the scientific name of the daffodil, it must be italicized as it is the genus name of daffodils. I would like to recommend to use the English name.

Secondly the "melting pot of potyviruses" title part suggest that the authors investigated all of the 8 formerly described potyviruses notedly: Cyrtanthus elatus virus A (syn. Vallota speciosa virus), Hippeastrum mosaic virus, Lily mottle virus, Narcissus degeneration virus (NDV) (syn. chinese narcissus potyvirus, Lycoris potyvirus), Narcissus late season yellows virus (NLSYV), Narcissus yellow stripe virus (Narcissus white stripe associated virus), Onion yellow dwarf virus and Ornithogalum mosaic virus. Instead, they investigated NLSYV and NDV in great detail.

From line 34 to 38 the authors write: "For instance, cyrtanthus elatus potyvirus A (CyEVA), iris severe mosaic potyvirus (ISMV), lily mottle potyvirus, narcissus late season yellows potyvirus (NLSYV), narcissus latent macluravirus (NLV), narcissus mosaic potexvirus, narcissus tip necrosis carmovirus (tentative member), narcissus yellow stripe potyvirus (NYSV), narcissus degeneration potyvirus (NDV), and ornithogalum mosaic potyvirus" There are obsolete virus names! I kindly ask the authors to use the currently accepted nomenclature of the ICTV which is available as part of the latest Master Species list: https://talk.ictvonline.org/files/master-species-lists/m/msl/12314

They used obsolete virus names in lines 40 to 44. In this paragraph the authors listed 16 virus species, but till today there have been 26 virus species and 3 tentative species reported from daffodils, either they list all reported species or list only all described potyviruses.

Line 54: “Narcissus plants are likely to harbor co-infections of potyviruses and other viruses in several areas of China, India, New Zealand, European countries, and Japan [13,22,23,37,38]…” These citations refer to Japanese, Chinese and New Zealander isolates and not from India and Europe. Please complete the references list with Indian and European data e. g. potyvirus infection on daffodils. (for example:

Chandel, Vanita, Singh, Manoj K., Hallan, Vipin and Zaidi, Aijaz A.(2010) 'Evidence for the occurrence of a distinct potyvirus on naturally growing Narcissus tazetta', Archives of Phytopathology and Plant Protection, 43: 3, 209 — 214,

Rashmi Raj, Charanjeet Kaur, Ashish Srivastava, Susheel Kumar, S. K. Raj. (2020) Sequence analyses of RT-PCR products obtained from seven infected leaf samples revealed existence of three potyvirus species in Indian narcissus (Narcissus tazetta L.). 3 Biotech 10:10.

Agoston, J., Almasi, A., Nemes, K., Salanki, K. and Palkovics, L. (2020) First report of hippeastrum mosaic virus, narcissus late season yellows virus, narcissus latent virus and narcissus mosaic virus in daffodils from Hungary. J Plant Pathol 102, 1275–1276

Dariusz Sochacki and Ewa Chojnowska. (2016) The Frequency of Viral Infections on Two Narcissus Plantations in Central Poland. Journal of Horticultural Research 24:2, pages 19-24.

In lines 59-60 they refer to "... the term NYSV-like virus refers to the narcissus viruses in the turnip mosaic virus (TuMV) phylogenetic group ..." For clarity, please list all potyviruses from the TuMV clade here.

In the Introduction part the authors may want to give more background of NLSYV and NDV. There is a great deal of information on NLSYV in Mowat, W.P.; Duncan, G.H.; Dawson, S. Narcissus Late Season Yellows Potyvirus - Symptoms, Properties and Serological Detection. Annals of Applied Biology 1988, 113, 531–544, doi:10.1111/j.1744-7348.1988.tb03330.x.

In the Materials and Methods part, the authors describe the RT-PCR protocol, but in scientific publications a plant is only considered infected with a pathogen if the authors detect it with at least 2 different methods, one of the methods that provides sequence evidence (e. g. RT-PCR and ELISA or Hybridization or NGS). These requirements are generally accepted and considered to be a "Gold Standard".

In line 87-88 ("...and other some potyviruses were amplified from narcissus leaves...") the sentence should be rephrased.

In lines 89-93 the authors describe the primes, but please also indicate the direction of the primers, which one is the forward and which is the reverse. Please also note what the underlined parts mean.

In lines 93-94 for clarity please list which viruses belong to the OYDV phylogenetic group.

Figure 1 and Table 1 should be in the results section as they both contain data obtained through this research. I also would like to suggest to make Figure 1 (map) as large as possible in the final version (and possibly make it enlargeable) as in this PDF the point and triangle marks are indistinguishable from each other.

In line 148 please clarify if nucleotide or amino acid sequences were used.

In line 152 please write out in full the name of JYMV, and put the acronym in parentheses.

In line 153 please write: accession number (acc #) and use only the abbreviated format acc # at all subsequent occurrence, this makes the text more readable and less redundant.

The authors mention "The alignments were made using CLUSTAL_X2...", please add the alignment in supplementary material (perhaps as a fasta or clustal file type).

In line 164 please clarify if nucleotide or amino acid sequences were used for the alignment. This alignment can be also added in the supplement.

In lines 183-187 the authors write they only used a part of the ORF of NLSYV, as this part did not contain recombination. I think that using the whole ORF in the phylogenetic analysis would make more sense, as this way the readers would clearly see how the sequences/accessions relate to each other with recombination, and then make 2 separate trees with the same method to see how they relate without/before recombination by one tree containing only the sections from the major parent and the other only the section of the minor parent. With these 3 trees the readers can compare and follow how the phylogenetic relationships change from the major and minor parents into the recombinant or vice versa. The parental trees (major and minor parents) are automatically generated by RDP, and one can copy the tree to the clipboard or to a file and save it.

In line 205: "The analyses were performed using default settings for the different detection programs..." The default setting for RDP v4.101 is sequences are circular. This setting in potyviruses can lead to incorrect recombination patterns and break points as these viruses have a linear genome, not circular.

In line 206: "...Bonferroni-corrected p-value cut-off of 0.05 or 0.01." Please clarify if the different cut-off values were used in different programs (e. g. RDP, GARD), or for different viruses (e. g. NLSYV, NDV), and also justify why the cut-off values were different.

In line 208: "...with an associated p-value of <1.0 × 10−4..." this seems to be in contradiction with line 206. Please clarify what was the cut-off value (0.005, 0.001 or 0.0001).

Similarly in line 210: "...100 and 50 nucleotide sliding window..." please explain why the sliding window length was different and which was used in which program or method.

In line 212 and 213 the authors describe that the sequences were aligned without outgroup and checked for evidence of recombination. The user manual of RDP does not recommend using alignments without an outgroup as this may lead to false positive recombination signals.

Line 214: "...GARD was used to assess sites for any evidence of recombination." While GARD is an alternative to RDP, GARD uses one method for recombination detection (genetic algorithm), RDP can use up to 8 different methods depending on the settings. Using GARD feels unnecessary when using RDP and only accepting recombinants when 3 different methods find it recombinant.

In Figure 2 authors may want to make it enlargeable to see the positions of the different accessions, or each clade for each virus could be enlarged and added as a supplement. It is very hard to interpret and match the accessions to the phylogenetic networks as the accessions are not visible on the tree. This is understandable as the authors investigated so many sequences. It would be clearer if there would be a table elaborating which accession belongs to which clade.

Line 341: these accessions are not yet available in GenBank. Please make them accessible. As a reviewer I was not able to verify if the accessions exist or if the support the findings of the authors.

Line 343: I suggest to rename the section: Inference of recombination. Both programs only infer the recombination events like phylogeny programs infer phylogenetic relations.

Line 347: Instead of data not shown please refer to Figure 2, which shows exactly this.

Line 352: Instead of "...unequivocal recombination sites..." please write recombination hotspots. It is not unequivocal as these inferred recombination sites have not been confirmed by in vivo or in vitro studies. Recombination hotspots denotes motifs where the program infers that recombinations are statistically more likely to occur in the analysed alignment. Please also clarify which program gave this result and provide supporting statistics in a table.

Lines 353 and 354: "...(genome position around 2000 nt)..." GARD and RDP provides inferred recombination breakpoints, RDP with a confidence intervals. Many methods in RDP are also able to give an estimated breakpoint. If 2 or more methods gave the same result it is highly probable that it is actually a recombination breakpoint. Also, RDP provides probability statistics for each recombination event for each used method how likely is this event caused by chance. The lower this probability is the more likely that this breakpoint is real. Without statistics I do not feel that this result is well supported.

Line 355 and 356: based on Japanese and Australian isolates one cannot conclude that "the site is probably distributed worldwide."

Line 357: "...positions around 9000 nt..." Once again, both GARD and RDP provides breakpoints, RDP with a confidence intervals and with probability statistics.

Line 358: "...NLSYV; however, the parental sequences are different." This needs to be rephrased, if parental sequences are not different, they are the same isolate.

Lines 358-360: "Although no identical recombination type pattern of the genome was found for Japan and Australia, identical non-recombinants in China and Japan were found." This sentence also should be rephrased.

Figure 3: Non recombinant part: Group 3 not know should be elaborated or re-written to avoid confusion. Recombinant part: Group 4 and Group 3 text is missing from the second bar (NY-F1, NY-FK266, NY-HG25, NY-HR39, NY-OS1). Group 3 text missing from the bottom bar (NY-CB1).

Line 378: "Phylogenetic Relationships and Nucleotide Diversities Assessed by Polyprotein Coding Sequences" please write: Inference of Phylogenetic Relationships...

Lines 388-389: "...that the genetic diversities of NLSYV are higher than those of NDV in Japan." should be: in Japan the genetic diversity of NLSYV are higher than those of NDV. (To emphasize the findings are applicable to Japan)

Figure 4: Please use isolate codes and accession numbers in the tree, by using both the reader can clearly and promptly identify sequences.

Lines 410-414: "NYSV is a historical virus species..." this section would be more appropriate in the Introduction.

Line 415: "Since these viruses..." to avoid confusion please write: Since NYSV, NLSYV, CyEVA, NDV have narrow...

Line 416: "...are mostly limited to narcissus plants..." this statement is incorrect. Please write Amaryllidaceae host plants, as NLSYV have been reported from Sternbergia lutea, Clivia miniata, and NDV from Lycoris.

Line 436-438: this is a duplicated part, please delete "is possibly due to the different periods of introduction into Japan for NLSYV and NDV: NDV might have been more recently introduced into Japan than NLSYV."

Line 457: instead of "major narcissus" please write: important daffodil

Line 458-459: "The plant leaves showed chlorotic stripes, stripes, and mosaic symptoms one month post-inoculation." Daffodil leaves only expand in spring, cell division does not occur. Symptoms only appear in the next growing season after sap inoculation. This has been described in spring flowering bulbs by several authors:

Dekker, E.L.; Derks, A.F.L.M.; Asjes, C.J.; Lemmers, M.E.C.; Bol, J.F.; Langeveld, S.A. Characterization of Potyviruses from Tulip and Lily Which Cause Flower-Breaking. Journal of General Virology 1993, 74, 881–887, doi:10.1099/0022-1317-74-5-881.

Mowat, W.P.; Duncan, G.H.; Dawson, S. Narcissus Late Season Yellows Potyvirus - Symptoms, Properties and Serological Detection. Annals of Applied Biology 1988, 113, 531–544, doi:10.1111/j.1744-7348.1988.tb03330.x.

Valverde, R.A.; Sabanadzovic, S.; Hammond, J. Viruses That Enhance the Aesthetics of Some Ornamental Plants: Beauty or Beast? Plant Disease 2012, 96, 600–611, doi:10.1094/PDIS-11-11-0928-FE.

Line 474: instead of "type patterns" please write: hot spots

Line 480: instead of "a species of wild onion" please add the scientific name of the plant. The English, Japanese name and wild onion makes it very hard to scientifically pinpoint what species the authors mention.

Lines 487-488: this whole sentence needs a citation and can go to Introduction. "The Japanese narcissus plant (Nihon-zuisen, Narcissus tazetta var. chinensis) was first described in the kanji dictionary as Kagakushu, which was published in 1444 CE in Japan."

Lines 489-490: "It seems likely that it drifted along oceanic currents from Chinese continent to Japan before the Muromachi period (1336–1573 CE) at least three times." Please provide scientific evidence with citation, because it is highly unlikely that daffodil bulbs would "drift" along oceanic currents. If there is none, please re-write this sentence: Japanese daffodils could have been introduced to Japan through trade with China during the Muromachi period (1336-1573 CE) [citation].

Line 491: "...Silk Road to the Eastern..." THE is not needed, please delete

Lines 496-497: "For Japanese garlic plants, there is no report of domesticated plants and no information on their introduction to Japan." please write: Japanese garlic plants are considered to be naturally part of the country's flora.

Lines 497-500: "The different frequencies of the coinfection between narcissus viruses and ScaMV might be related to whether they have a long history after emergence, trade history between countries, or migration history of invasive plants around the globe." This is pure speculation, this sentence should be re-phrased or omitted.

Lines 503-505: "Our findings also illustrate that it is important to understand the co-infections of viruses in nature when designing control strategies, as the wild and domesticated narcissus plants in Japan are somewhat like a melting pot of potyviruses and other viruses." In this experiment the authors collected samples from both wild and cultivated populations of daffodils. There were no mentions of control strategies or designing control strategies in the whole of the MS. Also, while daffodils known to be host of several virus species belonging to different virus genera, and to be infected with multiple virus species and isolates at the same time this does not mean in my understanding that they are a melting pot as there is no evidence of intraspecies recombination. In this paper the authors only wrote about the inference of recombination breakpoints of NLSYV, so I suggest this last paragraph should be deleted.

It would have been nice if the authors would compile a Discussion section - which is not mandatory, but helpful - where they discuss their results in the presence of already existing papers. I also would have appreciated if the authors would have mentioned/noted in the results section if their findings were in accordance or in contrast with earlier reports.

I think that by correcting the flaws of this MS it has the potential to become an interesting and valuable article in this journal.

Author Response

Comments and Suggestions for Authors

In this manuscript of an original article the authors collected 189 daffodil leaf samples of wild and domesticated plants in a 7-year period. They have found that 120 of the molecularly tested plants were infected with one or more members of the genus Potyvirus. Molecular testing, direct sequencing and later cloning and sequencing of this many samples is quite a job, tough in scientific circles it is expected that at least 2 different detection methods are used to verify the infection of a host. A possible second method could be ELISA testing, as it is relatively cheap and sensitive and can be upscaled, moreover there have been adequate number of reports using potyvirus group specific antibodies in bulbous ornamentals (including daffodils).

Response: Thank you for all your detailed comments. The comments were very useful to confirm our results again.

The title of the MS should be changed.

Firstly, Narcissus is the scientific name of the daffodil, it must be italicized as it is the genus name of daffodils. I would like to recommend to use the English name.

Secondly the "melting pot of potyviruses" title part suggest that the authors investigated all of the 8 formerly described potyviruses notedly: Cyrtanthus elatus virus A (syn. Vallota speciosa virus), Hippeastrum mosaic virus, Lily mottle virus, Narcissus degeneration virus (NDV) (syn. chinese narcissus potyvirus, Lycoris potyvirus), Narcissus late season yellows virus (NLSYV), Narcissus yellow stripe virus (Narcissus white stripe associated virus), Onion yellow dwarf virus and Ornithogalum mosaic virus. Instead, they investigated NLSYV and NDV in great detail.

Response: Thank you for all your comment. Daffodils (Narcissus pseudonarcissus) are native to northern/southern Europe and Japanese or Chinese Narcissus (Narcissus tazetta var. chinensis, Nihon-zuisen) is a little different from western daffodil. Therefore, we explained this shortly in the Introduction section. We also discussed this with our colleagues when we submitted our two articles a few years ago (IGE and PLoS One) and we decided to use Narcissus. Thus, we amended ‘narcisuss’ throughout the manuscript to ‘Narcisuss’.

Table 1 listed the viruses detected in this study. We used potyvirus-specific primer pairs which detect most of potyviruses (Gibbs and Mackenzie, 1997, Zheng et al., 2010, and Ohshima et al., 2016). Thus, we did not only investigate formerly described potyviruses but also unknown potyviruses. From our earlier results of the sequencing published in PLoS One (2018), we found a new virus of NV-1. Moreover, the Review 2 recommended to use this title, so we left the title as is.

From line 34 to 38 the authors write: "For instance, cyrtanthus elatus potyvirus A (CyEVA), iris severe mosaic potyvirus (ISMV), lily mottle potyvirus, narcissus late season yellows potyvirus (NLSYV), narcissus latent macluravirus (NLV), narcissus mosaic potexvirus, narcissus tip necrosis carmovirus (tentative member), narcissus yellow stripe potyvirus (NYSV), narcissus degeneration potyvirus (NDV), and ornithogalum mosaic potyvirus" There are obsolete virus names! I kindly ask the authors to use the currently accepted nomenclature of the ICTV which is available as part of the latest Master Species list: https://talk.ictvonline.org/files/master-species-lists/m/msl/12314

Response: We amended accordingly, for instance, changed from ‘cyrtanthus elatus potyvirus A’ to ‘cyrtanthus elatus virus A’. The provided URL is not updated one, and the latest one is Potyviridae - Potyviridae - Positive-sense RNA Viruses - ICTV (ictvonline.org)

They used obsolete virus names in lines 40 to 44. In this paragraph the authors listed 16 virus species, but till today there have been 26 virus species and 3 tentative species reported from daffodils, either they list all reported species or list only all described potyviruses.

Response: Listed potyviruses and a macluravirus as suggested. Furthermore, we added ‘To date, 26 virus species and 3 tentative species were also reported to infect Narcissus plants’.

Line 54: “Narcissus plants are likely to harbor co-infections of potyviruses and other viruses in several areas of China, India, New Zealand, European countries, and Japan [13,22,23,37,38]…” These citations refer to Japanese, Chinese and New Zealander isolates and not from India and Europe. Please complete the references list with Indian and European data e. g. potyvirus infection on daffodils. (for example:

Response: Added the references below as suggested.

Chandel, Vanita, Singh, Manoj K., Hallan, Vipin and Zaidi, Aijaz A. (2010) 'Evidence for the occurrence of a distinct potyvirus on naturally growing Narcissus tazetta', Archives of Phytopathology and Plant Protection, 43: 3, 209-214,

Rashmi Raj, Charanjeet Kaur, Ashish Srivastava, Susheel Kumar, S. K. Raj. (2020) Sequence analyses of RT-PCR products obtained from seven infected leaf samples revealed existence of three potyvirus species in Indian narcissus (Narcissus tazetta L.). 3 Biotech 10:10.

Agoston, J., Almasi, A., Nemes, K., Salanki, K. and Palkovics, L. (2020) First report of hippeastrum mosaic virus, narcissus late season yellows virus, narcissus latent virus and narcissus mosaic virus in daffodils from Hungary. J Plant Pathol 102, 1275-1276

Dariusz Sochacki and Ewa Chojnowska. (2016) The Frequency of Viral Infections on Two Narcissus Plantations in Central Poland. Journal of Horticultural Research 24:2, pages 19-24.

In lines 59-60 they refer to "... the term NYSV-like virus refers to the narcissus viruses in the turnip mosaic virus (TuMV) phylogenetic group ..." For clarity, please list all potyviruses from the TuMV clade here.

Response: Added as suggested.

In the Introduction part the authors may want to give more background of NLSYV and NDV. There is a great deal of information on NLSYV in Mowat, W.P.; Duncan, G.H.; Dawson, S. Narcissus Late Season Yellows Potyvirus - Symptoms, Properties and Serological Detection. Annals of Applied Biology 1988, 113, 531–544, doi:10.1111/j.1744-7348.1988.tb03330.x.

Response: Added more background of NLSYV as suggested and cited Mowat et al. (1988) here.

In the Materials and Methods part, the authors describe the RT-PCR protocol, but in scientific publications a plant is only considered infected with a pathogen if the authors detect it with at least 2 different methods, one of the methods that provides sequence evidence (e. g. RT-PCR and ELISA or Hybridization or NGS). These requirements are generally accepted and considered to be a "Gold Standard".

Response: Thank you for your comments. Our main objective of this study is to find variants of NLSYV and NDV. Please think about the detection of SARS-CoV-2 variants. They were detected by RT-PCR and then sequence them to identify variants. We also believe this combination is the best to find variants. The immuno-detection is only used when they need a high speed detection but roughly. Moreover, in case of the plants were singly infected with a virus, the strategy proposed is apparently useful. However, when we need to differentiate the different viruses, different isolates of the same viruses and quasispecies (mutant clouds) of the same virus, ELISA and hybridization are less useful. It is known that NGS is still low sensitive compared to the PCR detection. It is said that the narcisuss virus antisera to one virus is cross-reactive to others.

In line 87-88 ("...and other some potyviruses were amplified from narcissus leaves...") the sentence should be rephrased.

Response: Rephrased as suggested.

In lines 89-93 the authors describe the primes, but please also indicate the direction of the primers, which one is the forward and which is the reverse. Please also note what the underlined parts mean.

Response: Done as suggested.

In lines 93-94 for clarity please list which viruses belong to the OYDV phylogenetic group.

Response: Clarified as suggested.

Figure 1 and Table 1 should be in the results section as they both contain data obtained through this research. I also would like to suggest to make Figure 1 (map) as large as possible in the final version (and possibly make it enlargeable) as in this PDF the point and triangle marks are indistinguishable from each other.

Response: Moved Figure 1 and Table 1 to the Result section. Amended triangles and enlarged the figure as suggested.

In line 148 please clarify if nucleotide or amino acid sequences were used.

Response: Clarified as suggested.

In line 152 please write out in full the name of JYMV, and put the acronym in parentheses.

Response: Added full the name of JYMV in the Introduction section.

In line 153 please write: accession number (acc #) and use only the abbreviated format acc # at all subsequent occurrence, this makes the text more readable and less redundant.

Response: Changed as suggested.

The authors mention "The alignments were made using CLUSTAL_X2...", please add the alignment in supplementary material (perhaps as a fasta or clustal file type).

Response: We sometimes see the alignments in very old papers, but have never seen such alignments in the recent papers. Everybody can now easily align the sequences using CLUSTAL_X2, MEGA and the automatic analysis give us same results, thus we decided that it was unnecessary to add the alignment.

In line 164 please clarify if nucleotide or amino acid sequences were used for the alignment. This alignment can be also added in the supplement.

Response: We didn’t add the alignments because of the reason above. However, we clarified as suggested,

In lines 183-187 the authors write they only used a part of the ORF of NLSYV, as this part did not contain recombination. I think that using the whole ORF in the phylogenetic analysis would make more sense, as this way the readers would clearly see how the sequences/accessions relate to each other with recombination, and then make 2 separate trees with the same method to see how they relate without/before recombination by one tree containing only the sections from the major parent and the other only the section of the minor parent. With these 3 trees the readers can compare and follow how the phylogenetic relationships change from the major and minor parents into the recombinant or vice versa. The parental trees (major and minor parents) are automatically generated by RDP, and one can copy the tree to the clipboard or to a file and save it.

Response: Thank you for your fine comments. We have confirmed recombination sites using the phylogenetic analyses as suggested, and added the ML trees in Figure S2. Sorry, we have found a mistake in recombination map (Figure 4). We have done similar approach in our earlier reports for TuMV and potyviruses published in JGV, Virus Research, PLoS ONE, Scientific Reports, Virus Evolution, FrontMicro, and PNAS. Our main studies were to identify recombination events for more than two decades.

In line 205: "The analyses were performed using default settings for the different detection programs..." The default setting for RDP v4.101 is sequences are circular. This setting in potyviruses can lead to incorrect recombination patterns and break points as these viruses have a linear genome, not circular.

Response: Described as suggested. Generally, the default setting includes circular and linear, and nobody describe ‘linear’ because this is a very minor setting.

In line 206: "...Bonferroni-corrected p-value cut-off of 0.05 or 0.01." Please clarify if the different cut-off values were used in different programs (e. g. RDP, GARD), or for different viruses (e. g. NLSYV, NDV), and also justify why the cut-off values were different.

Response: We have done both in RDP4 software, but listed p-values by cut-off 0.01, so we discarded 0.05, as suggested.

In line 208: "...with an associated p-value of <1.0 × 10−4..." this seems to be in contradiction with line 206. Please clarify what was the cut-off value (0.005, 0.001 or 0.0001).

Response: The p-value cut-off of 0.01 was just the setting that RDP4 shows as the possible recombination events, so the detected recombination events should be evaluated manually one by one. The p-value < 1.0 x 10-4 was the actual criteria that the recombination event was considered as exceeding the threshold.

Similarly in line 210: "...100 and 50 nucleotide sliding window..." please explain why the sliding window length was different and which was used in which program or method.

Response: Explained as suggested.

In line 212 and 213 the authors describe that the sequences were aligned without outgroup and checked for evidence of recombination. The user manual of RDP does not recommend using alignments without an outgroup as this may lead to false positive recombination signals.

Response: This is case by case. In case of different lengths of genomic sequences between outgroup taxa and main virus sequences, it is better to use the alignments with/without outgroup taxa to confirm the recombination events.

Line 214: "...GARD was used to assess sites for any evidence of recombination." While GARD is an alternative to RDP, GARD uses one method for recombination detection (genetic algorithm), RDP can use up to 8 different methods depending on the settings. Using GARD feels unnecessary when using RDP and only accepting recombinants when 3 different methods find it recombinant.

Response: We believe to use recombination detecting programs as many as possible when evidence of recombination is unclear (temporal). In this study, we also used original SiScan version 2 program, and the results sometime tell us slightly different to the results by SiScan in RDP4. GARD is also useful when we want to visualize the possible recombination sites. RDP4 is very good software but also not almighty, and we believe using the multiple and original programs are the best.

In Figure 2 authors may want to make it enlargeable to see the positions of the different accessions, or each clade for each virus could be enlarged and added as a supplement. It is very hard to interpret and match the accessions to the phylogenetic networks as the accessions are not visible on the tree. This is understandable as the authors investigated so many sequences. It would be clearer if there would be a table elaborating which accession belongs to which clade.

Response: Thank you for your fine comments. We confirmed the networks and added dots of isolates and tables in Figure 2 as suggested. The objective of this networks using the short sequences was to select representative isolate in each clade for the subsequent full genomic sequencing.

Line 341: these accessions are not yet available in GenBank. Please make them accessible. As a reviewer I was not able to verify if the accessions exist or if the support the findings of the authors.

Response: There is only a description in the Instruction for Authors of Viruses; ‘Sequences should be submitted to only one database’. Just after the article is published, all accession numbers will be open to public automatically and immediately by GenBank/EMS/DDBJ database. This is general rule of opening sequences to public. Please confirm our sequences in the earlier articles.

Line 343: I suggest to rename the section: Inference of recombination. Both programs only infer the recombination events like phylogeny programs infer phylogenetic relations.

Response: Renamed as suggested.

Line 347: Instead of data not shown please refer to Figure 2, which shows exactly this.

Response: Changed as suggested.

Line 352: Instead of "...unequivocal recombination sites..." please write recombination hotspots. It is not unequivocal as these inferred recombination sites have not been confirmed by in vivo or in vitro studies. Recombination hotspots denotes motifs where the program infers that recombinations are statistically more likely to occur in the analysed alignment. Please also clarify which program gave this result and provide supporting statistics in a table.

Response: Discarded ‘unequivocal’ not to confuse.

Lines 353 and 354: "...(genome position around 2000 nt)..." GARD and RDP provides inferred recombination breakpoints, RDP with a confidence intervals. Many methods in RDP are also able to give an estimated breakpoint. If 2 or more methods gave the same result it is highly probable that it is actually a recombination breakpoint. Also, RDP provides probability statistics for each recombination event for each used method how likely is this event caused by chance. The lower this probability is the more likely that this breakpoint is real. Without statistics I do not feel that this result is well supported.

Response: Described clearly the recombination sites were supported by several programs in Table S4.

Line 355 and 356: based on Japanese and Australian isolates one cannot conclude that "the site is probably distributed worldwide."

Response: Discarded as suggested.

Line 357: "...positions around 9000 nt..." Once again, both GARD and RDP provides breakpoints, RDP with a confidence intervals and with probability statistics.

Response: See Table S4.

Line 358: "...NLSYV; however, the parental sequences are different." This needs to be rephrased, if parental sequences are not different, they are the same isolate.

Response: Rephrased as suggested.

Lines 358-360: "Although no identical recombination type pattern of the genome was found for Japan and Australia, identical non-recombinants in China and Japan were found." This sentence also should be rephrased.

Response: Rephrased as suggested.

Figure 3: Non recombinant part: Group 3 not know should be elaborated or re-written to avoid confusion. Recombinant part: Group 4 and Group 3 text is missing from the second bar (NY-F1, NY-FK266, NY-HG25, NY-HR39, NY-OS1). Group 3 text missing from the bottom bar (NY-CB1).

Response: Done and modified as suggested.

Line 378: "Phylogenetic Relationships and Nucleotide Diversities Assessed by Polyprotein Coding Sequences" please write: Inference of Phylogenetic Relationships...

Response: Changed as suggested.

Lines 388-389: "...that the genetic diversities of NLSYV are higher than those of NDV in Japan." should be: in Japan the genetic diversity of NLSYV are higher than those of NDV. (To emphasize the findings are applicable to Japan)

Response: Emphasized as suggested.

Figure 4: Please use isolate codes and accession numbers in the tree, by using both the reader can clearly and promptly identify sequences.

Response: Used as suggested.

Lines 410-414: "NYSV is a historical virus species..." this section would be more appropriate in the Introduction.

Response: Deleted ‘NYSV is a historical virus species...’ for the clarity as suggested by one of reviewers.

Line 415: "Since these viruses..." to avoid confusion please write: Since NYSV, NLSYV, CyEVA, NDV have narrow...

Response: Changed as suggested.

Line 416: "...are mostly limited to narcissus plants..." this statement is incorrect. Please write Amaryllidaceae host plants, as NLSYV have been reported from Sternbergia lutea, Clivia miniata, and NDV from Lycoris.

Response: Changed as suggested.

Line 436-438: this is a duplicated part, please delete "is possibly due to the different periods of introduction into Japan for NLSYV and NDV: NDV might have been more recently introduced into Japan than NLSYV."

Response: Removed as suggested.

Line 457: instead of "major narcissus" please write: important daffodil

Response: Changed as suggested.

Line 458-459: "The plant leaves showed chlorotic stripes, stripes, and mosaic symptoms one month post-inoculation." Daffodil leaves only expand in spring, cell division does not occur. Symptoms only appear in the next growing season after sap inoculation. This has been described in spring flowering bulbs by several authors:

Response: We inoculated the viruses when the plant leaves were germinated and very small, namely before leaves expand in winter-spring season. We saw clear symptoms one month post-inoculation and confirmed the infections by RT-PCR after the leaves expanded. Described these in Materials and Methods section.

Dekker, E.L.; Derks, A.F.L.M.; Asjes, C.J.; Lemmers, M.E.C.; Bol, J.F.; Langeveld, S.A. Characterization of Potyviruses from Tulip and Lily Which Cause Flower-Breaking. Journal of General Virology 1993, 74, 881–887, doi:10.1099/0022-1317-74-5-881.

Mowat, W.P.; Duncan, G.H.; Dawson, S. Narcissus Late Season Yellows Potyvirus - Symptoms, Properties and Serological Detection. Annals of Applied Biology 1988, 113, 531–544, doi:10.1111/j.1744-7348.1988.tb03330.x.

Valverde, R.A.; Sabanadzovic, S.; Hammond, J. Viruses That Enhance the Aesthetics of Some Ornamental Plants: Beauty or Beast? Plant Disease 2012, 96, 600–611, doi:10.1094/PDIS-11-11-0928-FE.

Line 474: instead of "type patterns" please write: hot spots

Response: Recombination hot spots have different meaning, namely the regions where recombination have been frequently occurred in the viral genomes.

Line 480: instead of "a species of wild onion" please add the scientific name of the plant. The English, Japanese name and wild onion makes it very hard to scientifically pinpoint what species the authors mention.

Response: Added as suggested.

Lines 487-488: this whole sentence needs a citation and can go to Introduction. "The Japanese narcissus plant (Nihon-zuisen, Narcissus tazetta var. chinensis) was first described in the kanji dictionary as Kagakushu, which was published in 1444 CE in Japan."

Response: Moved to the Introduction section as suggested.

Lines 489-490: "It seems likely that it drifted along oceanic currents from Chinese continent to Japan before the Muromachi period (1336–1573 CE) at least three times." Please provide scientific evidence with citation, because it is highly unlikely that daffodil bulbs would "drift" along oceanic currents. If there is none, please re-write this sentence: Japanese daffodils could have been introduced to Japan through trade with China during the Muromachi period (1336-1573 CE) [citation].

Response: Rephrased and cited as suggested.

Line 491: "...Silk Road to the Eastern..." THE is not needed, please delete

Response: Deleted as suggested.

Lines 496-497: "For Japanese garlic plants, there is no report of domesticated plants and no information on their introduction to Japan." please write: Japanese garlic plants are considered to be naturally part of the country's flora.

Response: Rewrote as suggested.

Lines 497-500: "The different frequencies of the coinfection between narcissus viruses and ScaMV might be related to whether they have a long history after emergence, trade history between countries, or migration history of invasive plants around the globe." This is pure speculation, this sentence should be re-phrased or omitted.

Response: Discarded as suggested.

Lines 503-505: "Our findings also illustrate that it is important to understand the co-infections of viruses in nature when designing control strategies, as the wild and domesticated narcissus plants in Japan are somewhat like a melting pot of potyviruses and other viruses." In this experiment the authors collected samples from both wild and cultivated populations of daffodils. There were no mentions of control strategies or designing control strategies in the whole of the MS. Also, while daffodils known to be host of several virus species belonging to different virus genera, and to be infected with multiple virus species and isolates at the same time this does not mean in my understanding that they are a melting pot as there is no evidence of intraspecies recombination. In this paper the authors only wrote about the inference of recombination breakpoints of NLSYV, so I suggest this last paragraph should be deleted.

Response: We found co-infections of different viruses, different isolates of the same viruses and quasispecies (mutant clouds) of the same virus are common in narcissus plants, thus we used ‘a melting pot’ and this word does not include evidence of intraspecies recombination. One of the reviewers mentioned that the manuscript provides significant scientific results which are important to control the narcissus virus disease. Therefore, we made mention of potyviruses in the paragraph.

It would have been nice if the authors would compile a Discussion section - which is not mandatory, but helpful - where they discuss their results in the presence of already existing papers. I also would have appreciated if the authors would have mentioned/noted in the results section if their findings were in accordance or in contrast with earlier reports.

Response: We compiled the Discussion and Conclusion sections, as suggested by you and other reviewer.

I think that by correcting the flaws of this MS it has the potential to become an interesting and valuable article in this journal.

Response: Thank you for your useful comments and suggestions.

Reviewer 2 Report

This manuscript provides a detailed molecular diversity study focussed on two potyviruses found to widely infect narcissus plants in Japan, narcissus late season yellows virus and narcissus degeneration virus. The paper is generally clearly written, and the catchy title represents the contents well. The authors should address the following criticisms and revise their manuscript accordingly.

Major comments:

  1. In several sections of the manuscript, the authors claim that they have done “epidemiological analyses” (eg. abstract, line 8), however in view of this reviewer, epidemiology should cover more than the distribution of a pathogen, such as the role of vectors and the wider ecosystem leading to disease spread in time and space. The epidemiological studies were clearly not “detailed” (lines 8, 469).
  2. Mention in the introduction p51 that the large polyprotein is known to be post-translationally cleaved by virus-encoded proteases.
  3. Figure 1 is difficult to read due to small symbols; maybe a larger map can be added to supplementary materials.
  4. Table 1 font is too small and should be printed instead in landscape format. Delete “of” after “viruses”. Explain the third column of “plants detected (detected/examined, %)”. Unclear what 9(37.5) represents in this context. Also, numbers in ( ) in the Total are not defined. Use “0” instead of “None” in the table.
  5. Phylogenetic analysis of NLSYV polyprotein coding region shows 6 groups (3.7), whereas a similar analysis of CP region reveals 5 clades (3.3). This should be discussed in relation to virus evolution and predicted recombination.
  6. The statement in lines 435-436 is rather speculative; is there any evidence for potential introduction dates? Furthermore, the sentence has been accidentally duplicated in lines 436-438 and the duplication needs to be deleted.
  7. Section 3.9 heading should be changed to “Co-infection with carlaviruses” and lines 450-462 would be better place in the Materials & methods section, instead of Results & discussion.
  8. The paragraph in lines 478-502 does not belong into “Conclusions” and should be moved to the Discussion. This will make the conclusions short and to the point.

Minor points:

  1. L39-40: delete “As shown above” and revise sentence to “Many viruses that infect narcissus plants …”:
  2. L44: specify what location is meant here by “in a limited area” – UK, Japan ?
  3. L49: delete “molecule”
  4. L64: “…two appeared to be major viruses in Japan in an earlier study [28].”
  5. Section 2.1: mention how samples were stored before analysis
  6. L87: correct typo to “some other”
  7. L114-117: not bold
  8. L116-117: …a part of …(underlined) is unclear… and there are no underlined letters
  9. L123: delete “of” twice
  10. L161, 162: delete “in length”
  11. L165 “were inferred … [50].” Is the same sentence as in L188; delete one.
  12. L319: unclear wording: “from a single molecular RNA”
  13. L410: The intended meaning of “NYSV is a historical virus species” is unclear and should be reworded.
  14. L417: “However, to validate this hypothesis …”
  15. L456: correct typo to “carlavirus-specific”
  16. L459: delete second “stripes”
  17. L465: reword “probably commonly co-infected” to indicate that previous studies have shown such co-infection and the authors’ data show that this is also the case in the samples they analysed.
  18. L472: “nucleotide sequence identities”

Author Response

Comments and Suggestions for Authors

This manuscript provides a detailed molecular diversity study focussed on two potyviruses found to widely infect narcissus plants in Japan, narcissus late season yellows virus and narcissus degeneration virus. The paper is generally clearly written, and the catchy title represents the contents well. The authors should address the following criticisms and revise their manuscript accordingly.

Response: Thank you for all your fine comments and suggestions. All were useful. And thanks again for praising our title too.

Major comments:

  1. In several sections of the manuscript, the authors claim that they have done “epidemiological analyses” (eg. abstract, line 8), however in view of this reviewer, epidemiology should cover more than the distribution of a pathogen, such as the role of vectors and the wider ecosystem leading to disease spread in time and space. The epidemiological studies were clearly not “detailed” (lines 8, 469).

Response: Discarded epidemiological as suggested.

  1. Mention in the introduction p51 that the large polyprotein is known to be post-translationally cleaved by virus-encoded proteases.

Response: Added as suggested.

  1. Figure 1 is difficult to read due to small symbols; maybe a larger map can be added to supplementary materials.

Response: Enlarged Figure 1 as suggested by one of reviewers, so left this in main text.

  1. Table 1 font is too small and should be printed instead in landscape format. Delete “of” after “viruses”. Explain the third column of “plants detected (detected/examined, %)”. Unclear what 9(37.5) represents in this context. Also, numbers in ( ) in the Total are not defined. Use “0” instead of “None” in the table.

Response: Amended as suggested.

  1. Phylogenetic analysis of NLSYV polyprotein coding region shows 6 groups (3.7), whereas a similar analysis of CP region reveals 5 clades (3.3). This should be discussed in relation to virus evolution and predicted recombination.

Response: Sorry, our mistakes. Discussed as suggested.

  1. The statement in lines 435-436 is rather speculative; is there any evidence for potential introduction dates? Furthermore, the sentence has been accidentally duplicated in lines 436-438 and the duplication needs to be deleted.

Response: We added ‘, considering the difference of the expanding area and nucleotide diversities of two viruses collected in Japan.’ And removed the accidentally duplicated in lines.

  1. Section 3.9 heading should be changed to “Co-infection with carlaviruses” and lines 450-462 would be better place in the Materials & methods section, instead of Results & discussion.

Response: Moved as suggested.

  1. The paragraph in lines 478-502 does not belong into “Conclusions” and should be moved to the Discussion. This will make the conclusions short and to the point.

Response: Moved to Discussion section as suggested.

Minor points:

  1. L39-40: delete “As shown above” and revise sentence to “Many viruses that infect narcissus plants …”:

Response: Done as suggested.

  1. L44: specify what location is meant here by “in a limited area” – UK, Japan ?

Response: Clarified the sentence. ‘…. virus were also reported to infect narcissus but those were distributed in the limited countries of the world [29]’

  1. L49: delete “molecule”

Response: Deleted as suggested.

  1. L64: “…two appeared to be major viruses in Japan in an earlier study [28].”

Response: Thanks, changed as suggested.

  1. Section 2.1: mention how samples were stored before analysis

Response: Mentioned as suggested.

  1. L87: correct typo to “some other”

Response: Corrected as suggested.

  1. L114-117: not bold

Response: Reformatted as suggested.

  1. L116-117: …a part of …(underlined) is unclear… and there are no underlined letters

Response: Discarded ‘(underlined)’.

  1. L123: delete “of” twice

Response: Deleted as suggested.

  1. L161, 162: delete “in length”

Response: Deleted as suggested.

  1. L165 “were inferred … [50].” Is the same sentence as in L188; delete one.

Response: Removed the first one, thanks.

  1. L319: unclear wording: “from a single molecular RNA”

Response: Added ‘but not from difference RNA molecule of same virus’ for the clarity.

  1. L410: The intended meaning of “NYSV is a historical virus species” is unclear and should be reworded.

Response: Deleted ‘is a historical virus species and’ for the clarity.

  1. L417: “However, to validate this hypothesis …”

Response: Thanks, changed as suggested.

  1. L456: correct typo to “carlavirus-specific”

Response: Corrected as suggested.

  1. L459: delete second “stripes”

Response: Deleted as suggested.

  1. L465: reword “probably commonly co-infected” to indicate that previous studies have shown such co-infection and the authors’ data show that this is also the case in the samples they analysed.

Response: Reworded as suggested.

  1. L472: “nucleotide sequence identities”

Response: Corrected as suggested.

Reviewer 3 Report

The manuscript presented the epidemiology and evolution of major viruses that infect narcissus plants. The NLSYV and NDV were determinated as major viruses. Co-infection of viruses and recombined NLSYV isolates were also detected. The manuscript provides significant scientific results which are important to control the narcissus virus disease. Please see my comments in attached file.

Author Response

Comments and Suggestions for Authors

The manuscript presented the epidemiology and evolution of major viruses that infect narcissus plants. The NLSYV and NDV were determinated as major viruses. Co-infection of viruses and recombined NLSYV isolates were also detected. The manuscript provides significant scientific results which are important to control the narcissus virus disease. Please see my comments in attached file.

Response: Thank you for all your fine comments and suggestions. All were useful.

Page 1, line 27; stable

Response: Changed as suggested.

Page 2, line 81;

Response: Changed as suggested.

Page 3, line 114-116

Response: Formatted as suggested.

Page 5, line 131

Response: Deleted as suggested.

Page 10, line 475-476

Response: Deleted as suggested.

Page 10, line 485-486

Response: Deleted as suggested.

Round 2

Reviewer 1 Report

Authors have applied the requested changes and critiques to the MS, which I accept. I recommend publishing the corrected MS as a full original research article.